# Pulsed Power Applications for Protein Conformational Change and the Permeabilization of Agricultural Products

**DOI:** 10.3390/molecules26206288

**Published:** 2021-10-18

**Authors:** Koichi Takaki, Katsuyuki Takahashi, Alexis Guionet, Takayuki Ohshima

**Affiliations:** 1Faculty of Science and Engineering, Iwate University, Morioka 020-8551, Japan; ktaka@iwate-u.ac.jp; 2Agri-Innovation Center, Iwate University, Morioka 020-8550, Japan; guionet@iwate-u.ac.jp; 3Faculty of Science and Engineering, Gunma University, Kiryu 376-8515, Japan; tohshima@gunma-u.ac.jp

**Keywords:** pulsed power, pulse electric field, enzyme activity, protein conformational change activity, permeabilization, polyphenol extraction

## Abstract

Pulsed electric fields (PEFs), which are generated by pulsed power technologies, are being tested for their applicability in food processing through protein conformational change and the poration of cell membranes. In this article, enzyme activity change and the permeabilization of agricultural products using pulsed power technologies are reviewed as novel, nonthermal food processes. Compact pulsed power systems have been developed with repetitive operation and moderate output power for application in food processing. Firstly, the compact pulsed power systems for the enzyme activity change and permeabilization are outlined. Exposure to electric fields affects hydrogen bonds in the secondary and tertiary structures of proteins; as a result, the protein conformation is induced to be changed. The conformational change induces an activity change in enzymes such as α-amylase and peroxidase. Secondly, the conformational change in proteins and the induced protein functional change are reviewed. The permeabilization of agricultural products is caused through the poration of cell membranes by applying PEFs produced by pulsed discharges. The permeabilization of cell membranes can be used for the extraction of nutrients and health-promoting agents such as polyphenols and vitamins. The electrical poration can also be used as a pre-treatment for food drying and blanching processes. Finally, the permeabilization of cell membranes and its applications in food processing are reviewed.

## 1. Introduction

The supply of food toward consumers is an important issue that contributes to a sustainable society. The food industry has to respond to the demands of a growing population, both in terms of nutrition and consumer tastes, such as anti-aging and healthy food. These tasks have to be achieved with available resources and within regulatory requirements related to food safety. There are many technologies used in food processing to achieve an effective food supply; however, novel and innovative technologies are still required for emerging challenges of global food security, including safety and quality issues in the modern food supply chain.

The food processing applications of pulsed power technologies, such as intense pulsed electric fields and time-modulated discharge plasmas, have been studied worldwide, and interest in them among researches is growing rapidly [1]. The repetitive operation compact pulsed power generators with moderate peak power suitable for use in food processing applications have been developed by many researchers for temporal and spatial control of intense electric fields and discharge plasmas. Compact pulsed power generators suitable for use in the food processing applications are important because the applications are mainly based on biological effects of the electric field [2] and bio-chemical reactions of chemically active species in plasma [3].

Pulsed power is the technique of accumulating energy during a relatively long period of time and of releasing the accumulated energy in an extremely short period of time as a high-power pulse composed of a high voltage and a large current but moderately low energy, i.e., low time-averaged power [4]. When the pulsed high voltage is applied between arbitrarily shaped electrodes, intense pulsed electric fields (PEFs) are produced between the electrodes, which causes biological effects such as electroporation (formation of pores) of cell membranes [5] and conformational change in proteins [6]. These phenomena can be used in food proceedings such as drying, pasteurization (sterilization of bacteria), permeabilization and fermentation [1]. The poration of cell membranes by PEFs also contributes an improvement in the extraction of juice, nutritional agents and antioxidant metabolites, such as polyphenols from agricultural produce [7]. In the process of fermentation, a metabolic process of yeast and bacteria, activity can be controlled by exposure to intense PEFs [8,9]. When a high voltage is applied between electrodes over the discharge onset criterion, discharge plasmas are generated, which causes biological effects through bio-chemical reactions [10]. The discharge plasmas also induce UV radiation, an intense electric field in the vicinity of a discharge channel and shock waves, which also have different biological effects such as pasteurization and permeabilization [9].

For pulsed power generators used in food processing, it is important to design them with repetitive high-voltage outputs with an optimum amplitude of voltage and waveform shapes, in order to deliver moderate pulsed power into the biologic loads [4]. This energy flow can be based on simple circuits consisting of passive discrete resistive-inductive-capacitive elements, transformers (in a lumped constant circuit) or transmission lines (in a distributed constant circuit) and switches, which transfer the energy stored in the electric fields of capacitors or the magnetic fields of coils [11,12,13]. Firstly, pulsed power generation and electric field distribution generated by applying a high voltage between arbitrarily shaped electrodes are outlined in Section 2. The exposure to an electric field affects hydrogen bonds in the secondary and tertiary structures of proteins; as a result, the protein conformation is induced to be changed. The conformational change induces an activity change in enzymes such as α-amylase and peroxidase. Secondly, the conformational change in proteins and the induced protein functional change are described in Section 3. The permeabilization of agricultural products is caused through the poration of cell membranes by applying a PEF or shockwave produced by pulsed discharges. The permeabilization of cell membranes is used for extracting nutrients and health-promoting agents such as polyphenols and vitamins. These applications are described in Section 4. Finally, all applications are summarized in Section 5.

## 2. Pulsed Power Generators for Food Processing

Pulsed power is the technology of accumulating energy during a relatively long period of time and of releasing the accumulated energy in an extremely short period of time as a high-power pulse composed of a high voltage and a large current but moderately low energy, i.e., low time-averaged power. Compact pulsed power generators have been developed for applications in food processing through the poration of cell membranes or conformational change in proteins. These applications demand an intense electric field with a short rise time and an optimum pulse width for each application [4]. In order to meet these demands, many types of pulsed power generators have been developed and applied to the applications. Here, the generation of transient voltage (pulsed power) is outlined.

### 2.1. Basic Circuit for Pulsed Power

Figure 1 shows familiar circuits combining a capacitor of an inductor and switches. Capacitors and inductors (known as reactive elements) are used as primary energy storage elements that store electrical energy in the form of electric fields (0.5 *εE*^2^ (J/m^3^), where *εE*^2^ is the dielectric constant and *E* is the electric field strength) and magnetic fields (0.5 *μH*^2^ (J/m^3^), where *μ* is the magnetic permeability and *H* is the magnetic field strength), respectively.

In the capacitor–resistor circuit (capacitive energy storage system), shown as Figure 1a, the electrical energy, 0.5 *CV*_0_^2^ (*V*_0_ is the initial charging voltage), is stored in a capacitor and then transferred into a load resistor, *R*_L_, through a closing switch, S. The load voltage and current after closing the switch, S, are obtained as follows, using the continuity of current in the circuit, formulas (1) and (2):(1)$$v\left(t\right)=V_{0}\exp\left(-\frac{t}{R_{L}C}\right)$$
(2)$$i\left(t\right)=\frac{V_{0}}{R_{L}}\exp\left(-\frac{t}{R_{L}C}\right)$$
where *t* is the time after closing switch S. Therefore, the power is transferred from the energy store element into a load resistor, *R*_L_, as follows formula (3):(3)$$p\left(t\right)\;=\frac{V_{0}{}^{2}}{R_{L}}\exp\left(-\frac{2}{R_{L}C}t\right)$$

In the inductor–resistor circuit (inductive energy storage system), shown as Figure 1b, the magnetic energy, 0.5 *LI*_0_^2^ (*I*_0_ is the initial current in the inductor), is stored in an inductor and then transferred into a load resistor, *R*_L_, by opening switch *S*_1_ and closing switch *S*_2_. The load voltage and current after closing switch *S*_2_ are obtained as follows, using Kirchhoff’s voltage law, formulas (4) and (5):(4)$$v\left(t\right)\;=R_{L}I_{0}\exp\left(-\frac{t}{L/R_{L}}\right)$$
(5)$$i\left(t\right)\;=I_{0}\exp\left(-\frac{t}{L/R_{L}}\right)$$

Therefore, the power is transferred from the energy store element into a load resistor, *R*_L_, as follows formula (6):(6)$$p\left(t\right)=R_{L}I_{0}{}^{2}\exp\left(-\frac{2}{L/R_{L}}t\right)$$

The capacitive energy storage pulsed power supply has been used as a conventional pulsed power supply, called a capacitor bank system, for PEF treatment (Figure 2) [14]. The energy storage capacitor is charged by a high-voltage DC power supply from several hundred volts to several tens of kV. After charging the capacitor, the accumulated charge in the capacitor is discharged by closing a switch to the batch vessel which contains the materials to be treated. The intensity of the electric field can be controlled with the charging voltage and the gap length between the plate electrodes in the batch vessel. The pulse shape is expressed as double exponential functions and the pulse width is determined by the resistive component of the circuit and the capacitance of the energy storage capacitor.

For applications in the agriculture and food processing industries, compact and repetitive operation pulse sources are required. Semiconductor power switching devices, such as insulated gate bipolar transistors (IGBTs), a metal–oxide–semiconductor field-effect transistor (MOSFET) and a semiconductor opening switch (SOS) are generally used to drive the pulse modulator with a high repetition rate. The pulse voltage can be generated by direct switching of a high-voltage DC between falling and rising phases. Figure 3a shows a schematic circuit which consists of an AC/DC converter circuit, four H-bridge-connected IGBTs and a pulse transformer which is used for amplifying the voltage to 10 kV. The pulse width and the pulse repetition rate are controlled by timing the gate trigger of the semiconductor switching devices. Duty factor (ratio of on time per pulse cycle) can be also controlled by the gate trigger timing as shown in Figure 3b [15].

Figure 4 shows (a) the circuit diagram and (b) the typical output voltage of the inductive energy storage system pulsed power generator used to drive non-thermal plasma reactors. The electrical charges stored in the capacitor, *C*_1_, are transferred to the pulse transformer by sparking the gap switch. The primary voltage, *V*_1_, of the pulse transformer is amplified theoretically five times owing to the ratio of primary and secondary windings of the pulsed transformers. The amplified voltage is outputted as the secondary voltage of the pulse transformer and then applied to the inductor, *L* (2.5 μH), and capacitor *C*_2_ (1 nF). As the result, the current flows through the diode with LC oscillation. In this circuit, fast recovery diodes (voltage multiplier, K100UF) are employed as a semiconductor opening switch (SOS). After the current direction reverses with LC resonance, the electrical charges are injected into the SOS during 100 ns of reverse time. The SOS diode recovers after the charge-injection phase, and then the current is interrupted in very short time. As a result, a high voltage pulse is generated, similar to surge voltage (inductive voltage), by the short time circuit current interruption as follows formula (7) [16,17,18]:(7)$$v_{o}\left(t\right)\;=-v_{2}\left(t\right)+\frac{1}{C_{2}}\int i_{sos}dt+L\frac{di_{sos}}{dt}≅L\frac{di_{sos}}{dt}$$

The typical waveforms of the circuit current, capacitor voltage and output voltage are shown in Figure 4b. The pulse voltage with a 27 kV peak amplitude and a 60 ns width (FWHM) is obtained using the IES circuit. An IES circuit is classified as a type of voltage amplifier caused by a short time interruption of a circuit current using an opening switch. In IES, the timing of opening the switch to interrupt the current is important because an output voltage strongly depends on the amplitude of the current just prior to the interruption. In general, the opening switch is designed to operate at just prior to the peak of the circuit current [19,20].

Figure 5 shows a 13 kV silicon carbide (SiC)-MOSFET-driven compact inductive energy storage (IES) pulsed power generator [21]. The rising and falling voltage rates of the SiC-MOSFET were 157 and 129 kV/μs, respectively. The maximum current of the drain was 128 A. The minimum on resistance obtained was 1.07 Ω. For an IES circuit, the output voltage and pulse width (FWHM) were obtained as 31.4 kV and 55 ns, respectively, at a charging voltage of 1100 V, as shown in Figure 6 [21]. The maximum energy transfer efficiency was 50.2% at a load resistance of 2.5 kΩ. The SiC-MOSFET-driven IES circuit has excellent performance as a compact pulsed power system in many industrial applications, such as driving a corona plasma reactor for gas decomposition, water remediations, etc.

### 2.2. Cascade Connection for Voltage Multiplication

In general, intense high voltages such as several-hundred kV are sometimes demanded for industrial applications. However, there are many difficulties (e.g., spark gaps for switching off very high voltages, the increase in the physical size of the circuit elements, high DC voltages to charge capacitors, and suppressing corona discharges during the charging period) in making an extremely high voltage with one-stage. To overcome the above-mentioned difficulties, a new arrangement in which a number of capacitors were charged in parallel and then changing the connection in a series through the spark gap switches was developed by Marx in 1923 [12]. Figure 7 shows the schematic of a three-stage Marx generator. The three capacitors, *C*, are connected in parallel and are charged up to *V*_in_ through charging resistors, *R*. After charging the capacitors up, the lowest spark gap, GS1, is fired by the trigger ignitor, which is followed by the simultaneous breakdown of all remaining spark gap switches, GS2 and GS3. As a result, the connection of the capacitors changes from parallel to series; then, a voltage of −3*V*_in_ is supplied with a polarity opposite that of the charging voltage.

Figure 8a shows a basic circuit for single-stage impulse generators. The capacitor, *C*, is charged up with a DC power supply through a charging resistor, *R*_C_. After charging the capacitor up, the spark gap switch is turned on by firing the gap switch with an ignitor. The ignition time of the spark gap switch is much shorter than the front time (*T*_1_). After the gap switch is closed, the output voltage between the resistance, *R*, can be roughly expressed as shown in Figure 8b as ${\left(\frac{R}{L}\right)}^{2}-\frac{4}{LC}\gg 0$. The time constants for the rise and fall of the output voltage are roughly estimated as *L/R* and *RC*, respectively, under the conditions of ${\left(\frac{R}{L}\right)}^{2}-\frac{4}{LC}\gg 0$ [13]. Therefore, we can control the waveform by choosing values of resistance, *R*, capacitance, *C* and inductance, *L*.

Figure 9 shows the schematic and photograph of a Marx circuit for agricultural applications [22,23]. The Marx generator consists of four 0.22 μF capacitors, charging resistors (1 and 5 MΩ) and spark gap switches. The capacitors are charged up using a high-voltage DC power supply up to 12.5 kV. The charging time is approximately 10 s because of the output current limitation of the DC charging unit (power supply). When a spark gap switch is closed, the remaining gas switches are sequentially closed automatically and the connection of the capacitors is changed from parallel to series. The size and weight of the Marx generator are 1.0 m × 0.45 m × 0.45 m and 39.4 kg, respectively.

### 2.3. Pulse Compression and Forming

In general, the output voltage and current of simple pulsed power circuits show a double-exponential-shaped waveform with a pulse width of a microsecond scale as shown in Figure 8b. However, sometimes more short period pulses, such as a nanosecond scale, are required in the applications through a protein conformational change or direct stimuli inside the cell. Moreover, square-shaped pulses are sometimes demanded in applications through the poration of biological cell membranes. To meet these requirements, pulse compression and/or pulse-forming circuits are sometimes employed to be combined with a conventional pulsed power system.

A magnetic pulse compression (MPC) circuit was developed for driving high-power pulse gas lasers such as excimer lasers and has been modified for use in many applications, such as driving corona reactors for environmental applications and bio-medical applications. Figure 10a shows a schematic of an MPC-type pulse power generator. In the MPC, the magnetic switch is the key component with which to determine the pulse shape [24]. A saturable inductor, in which there is an inductor wound on a magnetic core, is commonly used as a magnetic switch in an MPC circuit [25]. After the capacitor, *C*_0_, is charged to *V*_C_, a semiconductor switching device is switched on. Since the saturable inductor *SI*_0_ has a large value in inductance, the current of the semiconductor switching device keeps a low value during the initiation phase. As the result, the switching loss calculated from the voltage and current in the switching device is minimized. When the capacitor, *C*_1_, is charged up to *nV*_C_, where *n* is the amplification factor of the pulse transformer, PT, the stored energy of *C*_1_ transfers to *C*_2_ through the saturable inductor, *SI*_1_. Following the energy transfer, the energy transfer from *C*_2_ to *C*_3_ occurs through *SI*_2_. The output voltage is also compressed by *SI*_3_. The rise time of the voltage decreases gradually because of *SI*_1_ > *SI*_2_ > *SI*_3_, as shown in Figure 10b.

Most applications for pulsed power require a constant voltage during period of pulse width. However, the output voltage is critically damped when using the pulsed power generator with a single capacitor and inductor. The output voltage waveforms can be changed from critically dumping (double-exponential-shaped) to square (constant in an arbitrary period) by pulsed power sources with multiple elements; such circuits are called pulse-forming networks (PFNs), whose transmission line (distributed constant circuit) is also used as pulse-forming lines (PFLs) [4,25].

Figure 11 shows a PFN circuit consisting of discrete elements of capacitors and inductors. PFNs can be analyzed using a finite number, *N*, of inductor–capacitor units instead of distributed inductor, *L*, and capacitor, *C*, components. The impedance of PFNs is obtained as $Z_{0}=\sqrt{L/C}$, where the quantities of *L* and *C* are the inductance and capacitance of discrete elements. *Z*_0_ is used to match with the load, i.e., a criterion of no reflection at the load. The traveling time of a voltage and current wave through a PFN is expressed as *N*$\sqrt{LC}$. Therefore, the output voltage pulse is obtained as an amplitude of *V*_0_/2 and a pulse duration of $\Delta t_{p}=2N\sqrt{LC\;}$at the connecting resistor of *Z*_0_ as a load. Figure 12 shows an example of a designed PFN and its output voltage by connecting a load resistance of *Z*_0_ = *Z*_L_ (matching condition; no reflection at the end of the PFN). The PFN consists of a 2 nF capacitor and 1.25 μH with *N* = 10. The charging voltage is 10 kV. The pulse length is calculated as $\Delta t_{p}=2\times 10\sqrt{1.25\times 10^{-6}\times 2\times 10^{-9}}=1\;\mu s.$ Figure 12b indicates that the pulse length and the amplitude of the output pulse are almost same values of calculating the results of 1 μs 5 kV (=*V*_0_/2).

PFNs are generally used in a time range longer than 1 μs, owing to the limitation of the *LC* value of discrete elements. For shorter pulse lengths lower than 1 μs, distributed element circuits, such as a transmission line (PFL; pulse-forming line), are commonly used to achieve better output waveforms. The capacitor and inductor elements of transmission lines have much smaller capabilities (capacitance and inductance) compared to discrete elements of capacitors and inductors. As a result, PFLs can generate square-shaped pulses in a range from 5 to 200 ns (5 ns < Δ*t_p_* < 200).

Figure 13 shows a three-stacked Blumlein line pulse generator using six coaxial cables as PFLs [26]. On the primary side, these lines are connected in parallel, but on the load side these lines are connected in series. This arrangement results in an output impedance of 300 Ω at a cable impedance of 50 Ω. The spark gap switch (SGS) is used as closing switch. The gap switch is controlled by triggering the ignitor. Figure 14 shows the typical output voltage of the stacked Blumlein line generator for different lengths of the coaxial cable (4, 6, 8, 10 and 12 m) with a corona discharge plasma reactor as a load. It can be confirmed that the pulse width of the output voltage changes when the cable length is changed [26].

## 3. Protein Conformational Change by PEF Irradiation

Intense PEFs are being tested for their potential uses in food processing, such as the inactivation of bacteria, enzyme activity control and fermentation acceleration, as non-thermal processing can have a reduced influence on food quality [1,9]. The advantage of PEF treatment compared to a thermal process is that it reduces detrimental changes in nutrition by retaining the physical and sensorial qualities of food [27]. PEF treatment is applied to a wide range of foods such as liquid (juice, milk, beer, etc.), semi-solid (gel-state foods) and solid-state foods. The typical operation range of PEF treatments is an electric field amplitude of 5–50 kV/cm with a pulse length in the range of several to tens of μs. PEFs are applied to food located between two electrodes and causes bacteria and enzymatic inactivation at a temperature lower than in thermal treatment [8,28]. The enzymatic reactions are a basic function of proteins, which are determined by its conformation of the polypeptide chain. Therefore, enzyme activity is affected by protein conformational change, such as the misfolding of proteins. The protein conformation consists of secondary structures (such as α-helices and β-sheets) and tertiary structures. The exposure of proteins to intense PEFs causes a conformational change through processes of electrical charging-up and the displacement of elements by electrical force [29]. In this section, the protein conformational change and enzyme inactivation by PEF irradiation are reviewed.

### 3.1. Conformational Change in Proteins

The exposure of proteins to intense PEFs causes conformational structural change through directly or indirectly affecting the secondary and tertiary structures. The direct effect of intense PEFs is a stretching of the molecular bindings in a protein, i.e., an unfolding structure caused by an electrostatic tensile force. Jiang et al. reported that the secondary structure was predicted to change from a helix into turns or random coils at an electric field with a strength of *E* > 0.5 V/nm (5.0 × 10^8^ V/m) using molecular dynamics (MD) calculations as shown in Figure 15 [30]. The MD calculations are carried out for the 1BBL (consisting of 37 amino acid residues) protein molecule that includes two α-helix secondary fragments. The original 1BBL protein structure still remains when there is no exposure to an electric field for all orientations during the period set in the calculation. However, the protein is stretched for all orientations with exposure to an electric field with relatively high strength. This result indicates that the realignment of some charged residues is induced by the exposure to an electric field. The structure change from an α-helix to the turns and random coils is caused to be more rapid by increasing the electric field strength.

Hydrogen bonds (HBs) are important in their role of stabilizing the conformation of, e.g., secondary structures. Jiang et al. also calculated the average number of total HBs in protein structures using MD simulation at various strengths of an exposed electric field, as shown in Figure 16 [30]. The average number of intra-protein HBs decreases with an increasing electric field strength larger than 0.5 V/nm. The number of intra-protein HBs has a strong relationship with the conformational structure stability of the protein. Moreover, the radius of gyration, *R*_g_, has the opposite tendency, against that of the number of intra-protein HBs.

Qin and Buehler reported that the protein secondary structural transitions depended on the amino acid chain length. The short amino chain proteins with fewer than 26 amino acids (i.e., 3.8 nm in length) are easily induced as interprotein sliding. However, the long amino chain proteins with larger length causes a conformational change from α-helix to β-sheet, which lead to increase the protein stiffness, strength, and energy dissipation capacity [31,32]. Valle et al. reported MD analysis of the conformational change of a single superoxide dismutase (SOD1) enzyme by exposing it to a 100-ns-wide intense PEF in the range of 10^8^ to 7 × 10^8^ V/m in strength [33,34]. In the MD calculations, a monopolar (MP) or a bipolar (BP) 100 ns PEF is applied to SOD1. The intensity of 7 × 10^8^ V/m induces a dramatic structural change with an irreversible transition from β-sheets or coil structures to unfolded states, as shown in Figure 17 [33].

Ding et al. calculated the electric force on the proteins which induces the conformational change with applied forces relative to the inter-chain bonding forces [35]. The inter-chain bonding of HBs in the α-helix and β-sheet was 8.1 kJ/mol (1.93 kcal/mol) and 6.6 kJ/mol (1.58 kcal/mol), respectively. Using the bonding energies of HBs and a distance between the elements of 0.35 nm, the inter-chain bonding forces of HB are obtained as 40 pN, which corresponds to approximately 10^8^ V/m in electric field strength. The transition in conformational structure from α-helices to β-structures was also analyzed based on the four-bead model using discrete MD modeling. The potential energy (εHB) of a β-hairpin structure is larger than that of an α-helix. However, the entropy of a β-hairpin is larger than that of an α-helix. From the free energy of the HB for α-helix and β-hairpin conformations, the α-helix-to-β-hairpin transition is predicted to be caused at 0.125 εHB of the temperature. Here, the connections of primary structures consist of covalent bonds such as peptide bonds and disulfide bonds (S–S). These bonds have almost one order higher bonding energy (210–630 kJ/mol). For this reason, the primary structure is generally less sensitive to electric fields compared to secondary and tertiary structures.

The conformational changes in proteins were also confirmed in relatively low electric field strengths (<0.5 V/nm) and exposure for long periods of time. Bekard and Dunstan reported conformational change lysozyme in an AC low electric field of 10 Hz in a frequency with a range from 0.78 to 5.0 V/cm, as shown in Figure 18 [29]. The conformational changes are monitored with the time evolution of the relative emission intensity of lysozyme solutions at 346 nm of the tryptophan fluorescence emission with an excitation wavelength of 295 nm. The conformation during the first hour is monitored without being exposed to an AC electric field, shown by dotted vertical lines, followed by 3 h with AC electric field exposure and a further 2 h without the electric field again. In the experiment, it was confirmed that the tryptophan fluorescence emission not only decreased its intensity, but that the red shift of the emission wavelength peak was caused by exposure to the electric field. The spectral changes generally indicate alterations in the microenvironment of tryptophan residues, and typically reflect the exposure of these residues, initially concealed in hydrophobic segments of the folded protein, to the surrounding aqueous environment. The decrements of the relative fluorescence emission intensity of lysozyme are observed for exposure to all electric field strengths, and is more pronounced at a field strength of up to 5.0 V/cm. The decrease in tryptophan emission intensity appeared irreversible. Further analysis of the data indicates a linear relation between the relative tryptophan emission intensity and the applied electric field strength, as shown in Figure 19 [29].

Bekard and Dunstan also reported that the fractions of the secondary structures of lysozyme solutions were changed from 31% α-helix, 20% β-strands, 20% β-turns and 29% random coils to 19% α-helix, 28% β-strands, 23% β-turns and 30% random coils after 3 h of exposure to an electric field of 3.0 V/cm strength. The electric field strength of 3.0 V/cm corresponds to be 0.1 fN of electrical force on HBs in the protein, which is almost six orders of magnitude lower than the HB bonding forces. To solve the inconsistency, Bekard and Dunstan proposed the model of indirect effect (slow process), which is based on the electrophoretic motion (electrostatic interactions) of a protein leading a frictional force for the protein unfolding [29]. The electrostatic effect is basically caused by oppositely charged terminal residues, charged side chains and peptide dipoles in the secondary structure segments of a protein. The dipole moment of lysozyme, which has a net charge of +7, is roughly calculated as 74 Debye length at natural pH. The alignment of secondary structure dipoles strongly affects the stability of the tertiary structure of proteins. In addition, the macro-dipole can distort the field distribution and produce relatively strong local electric fields. The electric field strength along a helix axis is estimated to be in the region of 10^9^ V/m.

### 3.2. PEF Treatment for α-Amylase Inactivation via Conformational Changes

In some food processes, such as brewing and fermenting, the inactivation of enzymes is the final step before distributing food products to consumers. In the processing of frozen food of agricultural products, hot water treatments are commonly used as blanching, which is used for inactivating microorganisms and enzymes at the final stage of the process. PEF treatment is one of the candidates used to alternate non-thermal methods for enzyme inactivation instead of the thermal process. The PEF treatments for enzyme inactivation have been investigated by some researchers [36]. Yeom and Zhang confirmed that the functions of enzymatic proteins were inactivated by PEF treatment in some optimized circuit parameters [37]. Vega-Mercado et al. also reported that PEF parameters such as strength, pulse width, number of pulses and rise time of the pulse mainly affected the efficiency of enzyme inactivation. They also confirmed that inactivation of enzymes required generally more energy (i.e., PEF strength, pulse width, and number of pulses) than microorganisms did [38]. Castro et al. indicated that the pulse width of a PEF was more important for the inactivation of enzymes than PEF strength was. They confirmed that the enzymic protein of alkaline phosphatase in milk was inactivated by 65% in a PEF of 22 kV/cm strength, 0.7 msec width and 70 pulses, whereas it was not inactivated in a PEF of 26 kV/cm, 0.39 msec and 20 pulses [39]. Concerning the inactivation mechanism of enzymes by the PEF treatment, Dong et al. pointed out that the conformational changes in enzymic proteins such as denaturation and aggregation caused the inactivation of enzymic proteins [40].

In basic experiments on enzyme inactivation by PEF treatment, small-scale vessels are sometimes used with parallel plane electrodes to generate homogeneous electric fields between the electrodes. Guionet et al. reported the effect of PEF treatment on enzymic inactivation of α-amylase. They developed a PFN circuit for controlling the pulse width and strength of a PEF, which was applied between the electrodes with a 4 mm gap in a cuvette, as shown in Figure 20 [6]. The cuvette was filled with α-amylase solution, which was prepared by dissolving 25 mg α-amylase in a solution consisting of 48 mL of distilled water and 2 mL of phosphate buffer. The results showed that the residual activity of α-amylase decreased with PEF strength at the same input energy with 10 μs of pulse width, as shown in Figure 21 [9]. This result indicates that the PEF strength strongly affects the efficiency of protein conformational change. They also confirmed conformational change in proteins due to PEF treatment, as shown in Figure 22 [9]. The tertiary structure change in α-amylase was monitored by fluorescence spectra at a 280 nm wavelength of excitation light. The tertiary structure (mainly tryptophan; Trp) of α-amylase also decreased with PEF strength at the same input energy. The enzymic active center of α-amylase was the carboxyl terminus of tryptophan. α-amylase commonly consists of three domains, which include several α-helix and β-sheet secondary structures. PEF treatments mainly affect hydrogen bonds in secondary structures (i.e., α-helix and β-sheet structures) and tertiary structures of α-amylase. It was also confirmed that the PEF and the heat treatments were different pathways for enzyme inactivation, as shown in Figure 23 [9]. They checked the aggregation of proteins after treatments by a PEF with 12.5 kV/cm and heating up to 70 °C. Both treatments caused the inactivation of α-amylase in same level of lower than 0.01 U/mL in residual activity. The relative protein concentration after filtering with a 0.22 μm syringe filter of PEF-treated α-amylase solution is almost same level as the control (without treatment), whereas the protein concentration of heat-treated α-amylase solution decreases to approximately 37%. The aggregation is confirmed to be only caused by heat treatment, because a decrease in the relative protein is caused by aggregation. Therefore, PEF treatment mainly contributes to conformational change in the protein, resulting in enzymatic activity change [6].

### 3.3. Enzyme Inactivation by a PEF under Various Conditions

Refolding denatured proteins has the potential to recover the enzymic activity of denatured enzymes, and this is an important issue for effective enzyme usage in the food industry [41]. PEF treatment induces the conformational change in proteins. Therefore, PEF treatment is a candidate for a novel method to refold the denatured enzymes. Ohshima et al. confirmed that the activity of six kinds of enzyme increased by a rate of 105%–120% via PEF treatment. They concluded that these enzyme activations were caused by protein conformational change or enzyme hydration [8]. They also reported that the activity of thermally denatured peroxidase recovered to 60% of its initial activity via PEF treatment with 12 kV/cm strength, 50 Hz repetition rate and 30 s exposure, whereas the enzymic activity recovered to only 40% of its initial activity at spontaneous refolding of the enzyme. However, the activity of thermally denatured lactate dehydrogenase (LDH) decreased due to PEF treatment, which suggested that further inactivation was caused by the application of a PEF to the thermally denatured LDH [8]. Therefore, the effect of PEF treatment on the refolding of the denatured proteins depends on the protein structure, i.e., the type of enzyme.

PEF treatment is a relatively low-temperature process compared to heat treatment, and is effective for the inactivation of not only foodborne and food spoilage bacteria but also enzymic proteins without degrading nutritional and sensory properties [42,43]. However, sometimes the PEF treatment alone for enzyme inactivation requires a long period of process time or significant electrical input power. Shamsi et al. proposed a combination of moderate heat treatment and PEF treatment to enhance the efficiency of inactivating enzymes and bacteria in whole milk [44]. Ho et al. confirmed that inactivating enzymes generally required more input energy in PEF treatment compared to the inactivation of microorganisms [45]. Agcam et al. conducted the inactivation of pectin methyl esterase (PME) in orange juice by PEF treatment. The inactivation of PME was significantly induced at a large input energy of PEF irradiation to the PME solution. A kinetic model was also proposed for estimating the efficiency of PME inactivation. In the model, the inactivation efficiency was expressed as a function of PEF treatment conditions, such as input power and treatment time. The kinetic model was confirmed to be effective for estimating the reaction rate and the time required for 90% inactivation [46]. Sharma et al. reported the effect of PEF treatment on the inactivation of four indigenous enzymes in whole milk. Lipase, plasmin, xanthine oxidase and alkaline phosphatase were used as specimens of indigenous enzymes in raw milk. The experimental results showed that the enzymic activities of plasmin, xanthine oxidase and lipolytic decreased with a 12%, 32%, and 82% reducing rate, respectively, compared to raw whole milk by a PEF treatment with 26.1 kV/cm strength at a 34 μs pulse width. When the strength of PEF increased to greater than 20.7 kV/cm for 34–101 μs, the enzymic activity of alkaline phosphatase was reduced to a comparable to thermal treatments. These results indicated that thermal effects also contribute to the inactivation of bacteria and enzymes along with the PEF treatments [47].

## 4. PEF Poration Process of Cell Membranes and Its Applications

Some applications of PEF treatment of biological cells in a conducting medium cause the charging up of the cell membrane, and the voltage across the membrane is then built-up. In case of the low electric fields, this voltage change induces gating, in which the opening of channels in the cell membrane is induced. An ion flux flowing through the ion channels causes a change in ion concentrations and balances in the vicinity of the cell membrane. This change in ion concentrations and balances works as a stress of cells. Stress for a short duration (in the order of milliseconds) and a small electric field do not cause irreparable damage. However, stress for a long duration and a high electric field causes damage as the permeability of the membrane increases to a level which results in either the recovery of cells jumping from seconds to hours (reversible breakdown) or cell death (irreversible breakdown) [48]. This section outlines the phenomena of cell membranes and their applications in food processing, such as the extraction of human-health-promoting agents and pre-treatment for improving the drying process.

### 4.1. Voltage Buildup across the Cell Membrane

Figure 24 shows a cross-sectional schematic of a biological cell and an equivalent circuit using a double shell model in suspension. The equivalent circuit consists of capacitive and resistive components [49,50]. The cell consists of cytoplasm, dissolved protein, electrolytes, glucose, nucleoplasm and other organelles. These components have relatively high conductivity. On the contrary, the membranes that surround the cell and subcellular structures have a low conductivity. Therefore, the cell can be thought of as a conductor (expressed as resistive components) surrounded by an insulating envelope which is expressed as capacitive components. These properties can be expressed as the equivalent circuit shown in Figure 24, in which the cell membrane is described by capacitance, *C*_m_, nuclear membrane by capacitance, *C*_n_, cytoplasm by resistances, *R*_2_ and *R*_4_ and nucleoplasm by resistance, *R*_3_, under the assumption that the conductance of the membranes is zero, and the capacitive components of the cytoplasm and nucleoplasm are negligible. Usually, capacitance *C*_m_ is higher than *C*_n_. Therefore, the applicability of the model is in a temporal range determined by the dielectric relaxation time of the membrane and cytoplasm.

The dielectric relaxation time, *τ*_r_, provides information on the impedance of resistive or capacitive components of the membrane and cytoplasm, respectively. *τ*_r_ is expressed as formula (8):(8)$$\tau_{r}=\varepsilon /\sigma ,$$
where *ε* is the permittivity and *σ* is the conductivity. For a pulse duration, *τ*, long compared to *τ*_r_, the resistive component is dominant, for the short to *τ*_r_, the capacitive component is dominant.

The amplitude of the critical voltage, *V*_crit_, across the membrane which affects the cell in aspects such as gating, poration or lysing depends on the cell type and its size as well as on pulse duration. The typical values of *V*_crit_ for lysing or poration are in the order of 1 V, for gating is approximately 100 mV [48].

The corresponding electric field *E*_crit_ in the medium (suspension) containing the cells is expressed as formula (9):(9)$$E_{\mathrm{crit}}=V_{\mathrm{crit}}/fa$$
where *a* is the cell radius and *f* is the form factor which depends on the cell shape. For example, the form factor, *f*, is obtained as 1.5 for spherical cells using void theory. In the case of cylindrical cells, the form factor, *f*, is expressed as formula (10):(10)$$f=l/\left(l-\frac{D}{3}\right)$$
where *l* is the cell length with hemispheres of diameter, *D*, at each end. The critical field strength, *E*_crit_, of bacteria with dimensions of 1 mm is estimated in the order of 10 kV/cm for lysing by the critical voltage, *V*_crit_, of 1 V for the pulse of tens of microsecond to millisecond durations. Microorganisms other than bacteria have dimensions in the range of 10–40 μm. Therefore, these microorganisms are much more vulnerable to electric fields compared to bacteria [48].

When a squared wave pulse with a voltage of *V*_a_ = *E* × *d* is applied, where *d* is the distance between electrodes and *E* is the electric field in medium, the voltage across the membrane, *v*_m_, at the poles increases with time, *t*, as follows formula (11):(11)$$v_{m}\left(t\right)=fE\left(D/2\right)\left[1-e^{-t/\tau_{c}}\right]+v_{0}$$
where *v*_0_ is the resting voltage (approximately 70 mV for many cells). The time constant for charging the membrane, *τ*_c_, is expressed as formula (12):(12)$$\tau_{c}=\left[\left(\frac{1+2V}{1-V}\right)\frac{\rho_{1}}{2}+\rho_{2}\right]C_{m}a$$
where *ρ*_1_ and *ρ*_2_ are the resistivities of the suspending medium and cytoplasm, respectively, *C*_m_ is the membrane capacitance per unit area, and *V* is the spherical cell’s volume. For a mammalian cell, the time constant for charging a cell membrane can be estimated as 75 nm using a 10 μm diameter, resistivities of 100 Ωcm and a volume concentration small compared to one (typical in vitro experimental conditions) [48].

### 4.2. Critical PEF Strength for Microorganism Survivability

The exposure of microorganisms to PEFs causes the charging up of the membrane, which induces damage as the permeability of the cell membrane increases to a level that either results in the recovery of cells increasing from seconds to hours (reversible breakdown) or cell death (irreversible breakdown) at PEFs with a long duration and high intensity. Therefore, the criterial E-field strength and exposure period for irreversible breakdown can be evaluated by the activity of the microorganisms.

The survivability of microorganisms, *s*, which is defined as the fraction of surviving microorganisms, decreases exponentially when increasing the amplitude of electric fields and linearly when increasing exposure time. The survivability, *s*, is expressed using an empirical law for a pulse of >50 μs duration and >8 kV/cm amplitude, as follows formula (13):(13)$$s=\left(\frac{\tau }{\tau_{0}}\right)\left(\frac{E-E_{\mathrm{crit}}}{E_{0}}\right)$$
where *τ* is the pulsed duration, *E* is the strength of the applied field, *E*_crit_ is the threshold field below which no effect is observed, and *τ*_0_ and *E*_0_ are constants which depend on the type and size of cell as well as the suspension medium, respectively. For example, Hülsheger obtained electrical parameters by analyzing the measured survivability, as shown in Figure 25, by best-fitting with *E*_crit_ = 4.9 kV/cm, *E*_0_ = 6.3 kV/cm, and *τ*_0_ = 12 μs [51]. From this empirical law, the required electrical energy density for lysing, *W*, is expressed as formula (14):(14)$$W=\tau \sigma E^{2}$$
where *E* is the electric field strength, *σ* is the conductivity, and *τ* is the pulse width. Equations (13) and (14) indicate that the high electric field and short pulse process serve to improve the efficiency of the process. Equation (14) also means that for long pulses thermal effects begin to play a role. For example, a 1 ms pulse with a voltage of 0.5 V across one membrane would lead to a temperature increase more than 20 degrees with the assumption of an adiabatic process. Therefore, the thermal effects also affect the process for a long width pulse.

The characteristic parameters *E*_crit_, *E*_0_, and *τ*_0_ for a short-duration pulse of <5 μs, however, differ from those for a long-duration pulse of >50 μs. For example, Schoenback obtained, with pulses of less than 2 μs duration, a change in *E*_crit_ to 40 kV/cm and a change in *E*_0_ and *τ*_0_ to 80 kV/cm and to 10 ns, respectively. The value of the electric field required for one-order reduction in *E. coli* in tap water is 164 kV/cm for 60 ns pulses, 107 kV/cm for 300 ns pulses, and 66 kV/cm for 2 μs pulses, as shown in Figure 26. The energies required for one-order reduction in *E. coli* are 0.85 J/cm^3^ for 300 ns pulses, 1.8 J/cm^3^ for 300 ns pulses, and 4.6 J/cm^3^ for 2 μs pulses [48]. Assuming that the biological processes are caused by intense, short pulses, the threshold of the electric field intensity, *E*_crit_, is given as formula (15):(15)$$E>E_{\mathrm{crit}}=\frac{V_{\mathrm{crit}}\left(2/fD\right)}{1-e^{\left(-t/\tau_{c}\right)}}$$

This equation can be written as formula (16):(16)$$E\tau >{\left(E\tau \right)}_{\mathrm{crit}}=V_{\mathrm{crit}}\left(2\tau_{c}/fD\right)$$

Using the equation, the critical applied electric field can be estimated as 13.3 kV/cm at a critical transmembrane voltage of 1 V, a spherical cell with a diameter of 10 μm, and a time constant of 100 ns for charging the membrane [49].

### 4.3. PEF Pasteurization

PEF treatments for the destruction of microorganisms, as pasteurization, have been investigated scientifically and practically since the early 1990s [52]. PEF pasteurization has advantages compared to conventional thermal pasteurization, such as fresh-like products and high nutritional quality. Influences of PEF treatment on the bioavailability of bioactive compounds contained in liquid foods were evaluated by some researchers and were summarized as review papers [53,54,55]. In particular, cow’s milk is one of most attractive liquid foods for PEF treatments, because the milk contains a high concentration of protein, in which the nutritional qualities are easily degraded by thermal treatment through the disnature of the protein. Therefore, the effects of PEF pasteurization on the quality of whole milk have been reported by some researchers [56,57,58].

Yang et al. confirmed the effect of PEF treatments as non-thermal processes on phenolic compound extraction from grapes in wine processing, and on the inactivation of spoilage microorganisms in wine, beer, and rice wine processing [7]. Sharma et al. reported the effect of a PEF treatment combined with a pre-heating process on microbial inactivation in whole milk. The pre-heating was controlled as 55 °C in temperature with a treatment period of 24 s, following stepwise cooling. The PEF treatment was set to be 22–28 kV/cm in strength with a 20 μs pulse width at a pulse repetition rate of 10–60 Hz. *Pseudomonas aeruginosa*, *Escherichia coli* (*E. coli*), *Staphylococcus aureus*, and *Listeria innocua* were used as specimens of microorganisms. The experimental results showed a 5–6 log reduction in all microorganism specimens to levels below detection limits [57]. Sharma et al. also reported reductions of 2–3 log in whole milk by PEF treatments with conditions of 20.7–26.2 kV/cm, 20 μs at 10–60 Hz. These microbial reduction levels of PEF treatment are almost equal to those of thermal pasteurization with 63 °C for 30 min (low-temperature pasteurization) or 73 °C for 15 s [47]. A typical PEF pasteurization system with a pre-heating unit for whole milk is shown in Figure 27 [59]. Ohshima et al. investigated using the PEF pasteurization system with 40 kV of voltage strength at a 50 Hz pulse repetition rate. Whole milk including *E. coli* was used as a specimen. The results showed that *E. coli* cells were not detected in treated milk. They confirmed that the processes of pre-heating and post-holding were effective for improving the efficiency of pasteurization [59].

There are many research papers on the effectiveness of PEF treatment in regard to the pasteurization of liquid food. However, thermal processes are still the primary method used in the food industry. For industrial applications of PEF pasteurization, the development and optimization of a PEF pasteurization system is necessary, including its power source, electrode configuration, pre-heat treatment, cooling unite, etc. For example, parallel plane electrodes have been commonly employed in PEF pasteurization because of the homogeneous electric field strength between the plane electrodes, i.e., the homogeneous effect on pasteurization. On the other hand, the parallel plane electrode configuration has some disadvantages, such as an accompanying large joule heating loss and pressure drop of liquid food flow in processing. Ohshima and Sato evaluated the energy efficiency in in PEF pasteurization for various configurations of electrodes. They employed parallel plane, needle-to-plane, ring-to-cylinder, and spiral winding configurations. The evaluation result showed that the PEF pasteurization efficiency depended strongly on the electrode configuration. The concentrated region in a non-uniform electric field was effective for the inactivation of microorganisms [60,61]. A novel textile electrode was tested for use in PEF processing by Kitajima et al. The textile electrode was combined with polyester fiber with tungsten wires at 0.2 mm in diameter. The effectiveness of the textile electrode was confirmed in PEF processing for the inactivation of *E. coli*. The inactivation efficiency was highest at 7 kV strength of applied voltage and showed a high value in low solution conductivity [62].

### 4.4. PEF Extraction of Intracellular Contents

PEF extraction of intracellular contents is based on the phenomenon of poration or the disruption of biological membranes through electromechanical compression processes by applying an intense PEF [63]. PEF extraction includes both reversible and irreversible disruptions of membrane, in which it is necessary to control the input energy into the cell membrane. The input energy from a PEF to the membrane is generally controlled by electrical parameters such as PEF intensity, repetition rate, and pulse width. The PEF treatment can be used in the extraction of health-promoting agents from vegetables and fruits.

Some researchers have confirmed that some intracellular contents, such as proteins including enzymes and waters, are extracted to the supernatant of cell suspension by applying a PEF to the specimens. Ohshima et al. confirmed the extraction of intracellular protein from yeast cells [64,65]. They also investigated the effect of cell membrane exposure to a PEF on extraction of intracellular protein and the recovery of target proteins using recombinant *E. coli*, as shown in Figure 28 [66]. The experimental results showed that the extraction efficiency of the target proteins (evaluated by enzymatic specified activity) depended on the electrical parameters of a PEF. The extraction selectivity of PEF extraction was higher than that of supersonic treatment. Shiina et al. reported that the recovery of extracted enzyme activities from a recombinant *E. coli* was improved by applying intermittent PEF treatment. The intermittent PEF was effective for the reversible disruption of cell membranes with a high survival rate of the cell [66]. They also reported an effect of PEF treatment on the production of α-amylase produced by recombinant E. coli during cultivation. The extracted α-amylase was approximately 30% of total α-amylase production, which was defined as the sum of extracellular and intracellular α-amylase in the recombinant E. coli, by applying a PEF of 12 kV/cm with intermittence as 50% duty (30 min on and 30 min off). Natural *E. coli* has no function with which to release secreted protein from intracellular to extracellular. However, PEF-assisted cultivation of *E. coli* enables the extracellular production of recombinant proteins [67].

Mahnič-Kalamiza et al. discussed the possibility of PEF extraction of health-promoting compounds from the residues of food processing, such as seeds, peels, grape husks, and oilcakes. They also evaluated the effect of PEF treatment on the extraction of variable compounds for biorefining agricultural and forestry residues such as stems, sawdust and leaves as well as bark, energy crops and municipal wastes, etc. [68]. This concept is almost the same as modern green technology, which is important for a sustainable food supply chain via renewable plant resources without agro-solvents (agro-chemicals). Barba et al. also discussed the effectiveness of PEF treatment on the extraction of valuable compounds from by-product wastes in food processing. They used biomass from terrestrial plants, energy crops, crop residues, forestry residues, grape pomace, food wastes, and beer waste brewing yeasts as specimens. These materials contain many bioactive compounds, especially polyphenols (such as phenolic acids, flavonol glycosides, anthocyanins, and catechins), which have functions of antiviral, antibacterial, antifungal, anticancer, and antioxidant effects. They also discussed the usage of PEF treatment for biorefinery applications [53].

Recently, PEF treatments were applied for the extraction of juices and nutritive molecules from agricultural products. Nakagawa et al. reported the effect of PEF treatment on polyphenol extraction from grape skins. The grape skins were immersed in distilled water between parallel plane electrodes. A PFN circuit was used to control the pulse width and intensity of PEF. The efficiency of total polyphenol extraction was evaluated as the gallic acid concentration in the solvent (water) by Folin–Ciocalteu analysis. The microscopic observation showed that membranes of anthocyanoplasts were raptured by PEF exposure, and the red-colored pigments in the anthocyanoplasts were extracted into the cell and solvent, as shown in Figure 29. The polyphenol extraction was enhanced with an increasing pulse width in the PEF treatment, i.e., the energy required for polyphenol extraction decreased with an increasing pulse width, as shown in Figure 30 [69].

### 4.5. PEF Pre-Treatment for Food Drying Processes

Drying is a food process in which water is removed to halt or delay the growth of microorganisms and chemical reactions responsible for producing spoilage [70]. Dehydration plays an important role in extending the shelf life of agricultural products. In addition to preservation, converting raw food into solid, dried food is an efficient way to reduce costs or relieve difficulties associated with packaging, handling, storage, and transport [71]. Hot-air drying is the most prevalent and one of the oldest methods of drying fruits and vegetable. Over 85% of industrial dryers are convective hot-air types [72]. However, hot-air drying has disadvantages, such as low energy efficiency, slow drying rates, and reductions in aroma, color, and nutrient contents [73].

Treatment with a PEF is an emerging non-thermal food processing method, in which an electric field is created between electrodes [1,5,9]. A PEF prevents an excessive temperature increase due to intermittent and short processing times [74,75,76]. A PEF causes electroporation [53] and improves the water permeability of cell membranes [77,78]. Barba et al. reviewed the current applications of PEFs in food science and industry. Their review mainly focuses on some of the most emerging PEF applications for the improvement of osmotic dehydration, extraction by solvent diffusion, or by pressing, as well as drying and freezing processes. The impact of PEF on different products of biological origin including plant tissues, suspension of cells, by-products, and wastes are analyzed in the review. In addition, recent examples of PEF-assisted biorefinery application are presented and PEF-assisted cold pasteurization of liquid foods is described [53]. Ade-Omowaye et al. also reviewed works on the use of pulsed electric fields as an upstream process in the dehydration and rehydration of plant-based foods. An effective and simple method for quantifying the extent of membrane permeabilization is discussed for the future and is highlighted [79].

Applying PEF can increase the subsequent drying rates of many kinds of agricultural products. Lebovka et al. reported the effect of PEF pre-treatment on the convective drying of potato tissue. The essential influence of PEF treatment at moderate electric field strengths (300–400 V/cm) on the drying of potato disks was that the effective moisture diffusivity increases with an increasing degree of PEF-induced damage, and that it is sensitive to the details of thermal pre-treatment procedures. For potato tissues, the PEF treatment allows the drying temperature to be decreased by approximately 20 °C; therefore, PEF pre-treatment is effective for enhancing the convective drying rate, especially for drying thermal-sensitive products at moderate temperatures [80]. Janositz et al. also reported an improvement in the cooking efficiency of PEF-treated potato slices. They used several-hundred width PEFs with a strength of 1.5–5.0 kV/cm for the treatment. They showed that the water loss of PEF-treated potato slices after baking in a convection oven increased with increasing PEF strength. Concerning oil uptake during deep-fat frying, PEF application to potato slices lead to a more effective reduction in fat content than in hot-water blanching [80]. Gachovska et al. reported the effectiveness of PEF pre-treatment during the drying and rehydration of carrots. A PEF intensity of 1 kV/cm or 1.5 kV/cm as well as blanched (100 °C, 3 min) carrots were used for the treatment. They showed that PEF pre-treatment increased the drying rate. There were no color differences between PEF-pretreated and blanched carrots before drying and after rehydration. In terms of texture, PEF-pretreated carrots were firmer than blanched carrots. PEF pre-treatment reduced the activity of peroxidase by 30–50% [81]. Lamanauskas et al. reported the effectiveness of PEF pre-treatment during the drying of kiwifruit (*Actinidia kolomikta*). The effectiveness of PEF pre-treatment was evaluated by a fluid bed hot-air dryer using PEF-pretreated *A. kolomikta* fruits. The result showed that the weight difference after PEF pre-treatment of the fruits was 13% after 3 h of drying, with a 5 kV/cm electric field strength, a pulse width of 20 μs, a pulse repetition rate of 20 Hz, and a total treatment time of 120 s. PEF pre-treatment did not influence the color parameters or ascorbic acid content of *A. kolomikta* fruits [82]. Telfser and Galindo reported the effect of reversible permeabilization as a pre-treatment before air-drying basil leaves at 40 °C, vacuum-drying, and freeze-drying them. The PEF pre-treatment shortened the drying times by 57% for air-drying, 33% for vacuum-drying, and 25% for freeze-drying. The influence of the PEF treatment on air-drying was also evident on tissue structures where the differences between the untreated and PEF-treated leaves were observed. Glandular trichomes on the surface of the leaves were better preserved when PEF was applied in combination with air-drying and vacuum-drying. PEF-treated, vacuum-dried samples were the closest to fresh leaves regarding color and smell according to the sensory panel [83].

In general, PEF treatment has been required to be soaked in water at the industrial level. The loss of water-soluble components by soaking in water is an important issue when producing high-quality products [79]. Yamada et al. tried to use PEF treatment without soaking in water to enable the treatment to prevent water-soluble components. They used a SiC-MOSFET pulsed power generator (PPG) to produce several-microsecond PEFs with 10 kV intensity as shown in Figure 31. The SiC-MOSFET PPG was compact and controlled the PEF strength and pulse width easily. They indicated that the PEF was effective as pre-treatment of air drying of various kinds of agricultural product such as eggplant, pumpkin, basil, radish, carrot, shiitake mushroom, kiwifruit, apple, scallops, wakame seaweed and spinach leaves [84]. Yamakage et al. also compared the impact of a pulsed electric field (PEF) with that of hot water (HW) and a control (CONT) on the subsequent drying rate, shrinkage, and quality parameters (degradation of *L-ascorbic acid*, *L-AsA*, and color) of spinach during hot-air drying. They also used a SiC-MOSFET PEF generator to produce PEFs with 1 μs in pulse width and several kV in strength. The PEFs were applied to spinach leaves that had not been soaked in water. PEF pre-treatment was conducted in 2.8 kV/cm of the electric field strength and 27.1 kJ/kg of the specific input energy. The drying rate for PEF samples was increased compared to CONT and HW samples as shown in Figure 32 and Table 1 [85]. The increase in the drying rate is due to the inhibition of shrinkage during drying as shown in Figure 33 and Table 2 [85]. Further, the degradation of *L-AsA* and the surface color of the PEF samples were significantly inhibited compared to HW samples. The PEF treatment could resolve the elution of water-soluble components caused by HW treatment [85]. Concerning PEF applications in the processing of spinach, Zhang et al. reported the extraction solution from spinach [77]. Yongguang et al. indicated the effects of PEF on protecting color in spinach puree [86]. This phenomenon is based on enzyme inactivation thorough the protein conformational change caused by PEF irradiation [87].

## 5. Conclusions

The applications of pulsed power technologies, including high electric field and time-modulated non-thermal plasmas, in food processing have been newly developed as non-thermal treatments, contributing to a modern food supply chain. Pulsed power technologies are used to produce time-modulated intense electric fields, which are employed to cause poration in biological cell membranes or conformational changes in proteins. These phenomena can be utilized in food processes such as extraction of health-promoting agents from agricultural products, pre-treatment of hot-air drying, inactivation of enzymes, and pasteurization.

The pulsed power technologies are based on temporally compressing energy. The direct capacitive discharge is commonly used as a simple pulsed power system with a single switch. An association of circuits to circumvent the still voltage and current limitation of semiconductors are also frequently used to generate an intense high voltage, such as an inductively multiplied circuit or a Marx generator. The pulse transmission lines and pulse-forming network are used to generate a square-shaped waveform transferred to the loads. The intense electric field can induce a biological effect such as electroporation.

In the food processing phase, an intense PEF can be used for non-thermal pasteurization via the phenomenon of electroporation. This phenomenon can be used to extract intracellular contents such as juice, nutritional agents, and health-promoting agents during food processes. PEFs can also contribute to enlarging the preservation period through the inactivation of enzymes and microorganisms based on protein conformational change.

Pulsed power applications in agriculture and food processing are new research fields and are still mainly in the experimental stage for developing industrial uses. To expand the applications from laboratories to industries, it is very important to develop an effective processing system including pre-treatment, a PEF treatment system, and optimal pulsed power generators for each application. For this issue, the collaborations between academia and industry are one of the key factors. In addition, clarifying the mechanism biologically and electrically for each application is also important to develop an effective system. For this issue, the collaborations among specialists and researchers in a variety of academic fields are also very important for the realization and development of these applications.

## Figures and Tables

**Figure 1 molecules-26-06288-f001:**
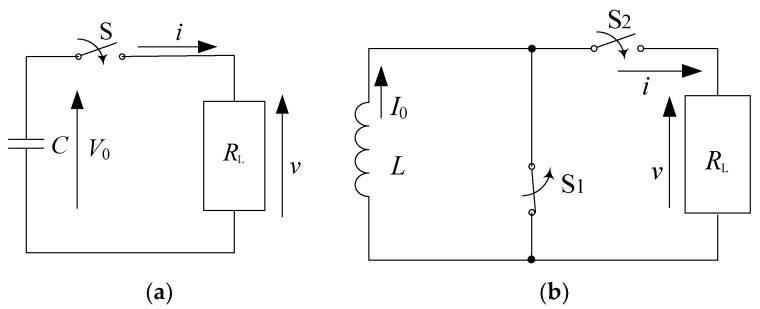
Basic circuits for pulsed power: (**a**) capacitive and (**b**) inductive energy storage systems. *C*: energy storage capacitor, *V*_0_: charging voltage, *S*: switch, *R*_L_: load resistor, *L*: energy storage inductor, *I*_0_: initial current, *S*_1_: opening switch, *S*_2_: closing switch.

**Figure 2 molecules-26-06288-f002:**
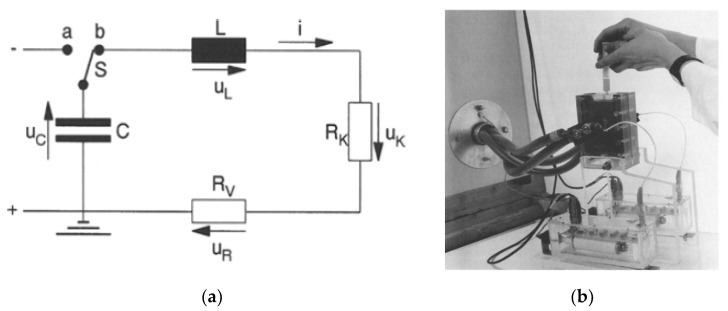
Circuit diagram of a PEF with a capacitor bank (**a**) and the batch vessel for sampling (**b**). C: energy storage capacitor, *L*: inductance, *R*_K_, *R*_V_: resistance, *u*_C_, *u*_L_, *u*_K_, *u*_R_: voltages of each device [14]. © Springer-Verlag 1996. With permission of Springer.

**Figure 3 molecules-26-06288-f003:**
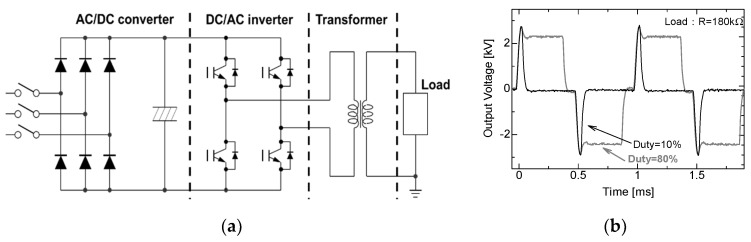
Schematic of (**a**) circuit consisting of an AC/DC converter, H-bridge with four IGBTs and a pulse transformer and (**b**) its output voltage waveforms [15]. © Walter de Gruyter 2016. With permission of Walter de Gruyter.

**Figure 4 molecules-26-06288-f004:**
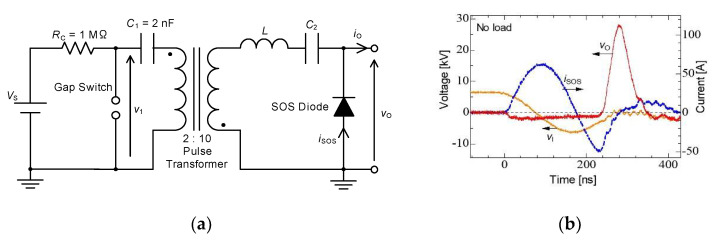
Basic circuits for pulsed power: (**a**) capacitive and (**b**) inductive energy storage systems. *V*_s_: DC high-voltage supply, *R*_C_: charging resistor, *C*_1_: energy storage capacitor, *L*: energy storage inductor, *C*_2_: secondary capacitor.

**Figure 5 molecules-26-06288-f005:**
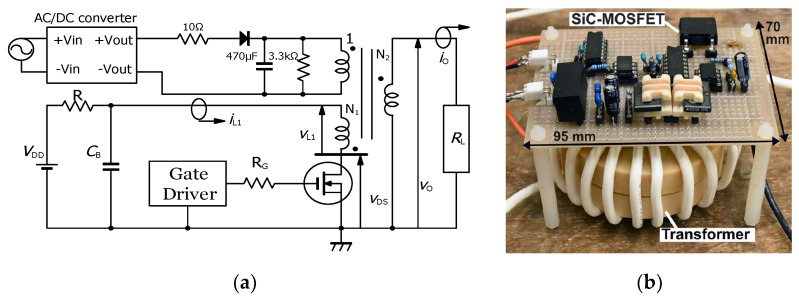
Circuit diagram (**a**) and photograph (**b**) of compact inductive energy storage pulsed power generator driven by a 13 kV SiC-MOSFET. *V*_DD_: DC power supply, *R*: charging resistor, *C*_B_: energy storage capacitor, *R*_G_: gate resistor, *R*_L_: load resistor [21]. © AIP Publishing 2021. With permission of AIP Publishing.

**Figure 6 molecules-26-06288-f006:**
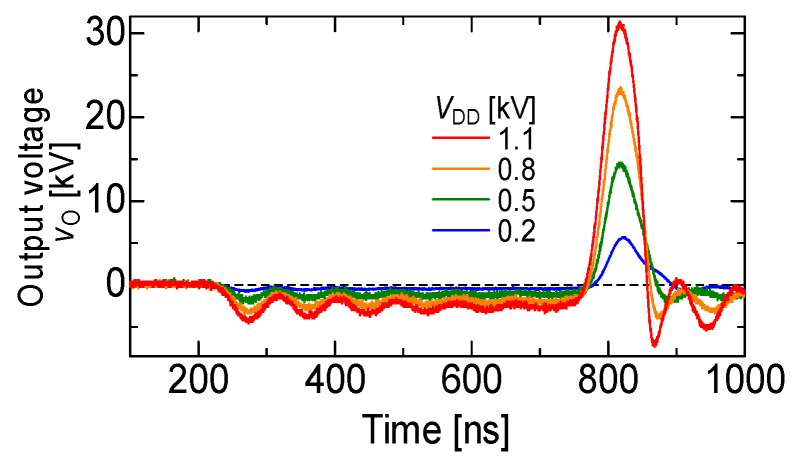
Waveforms of the output voltage of the secondary transformer for various input voltages [21]. © AIP Publishing 2021. With permission of AIP Publishing.

**Figure 7 molecules-26-06288-f007:**
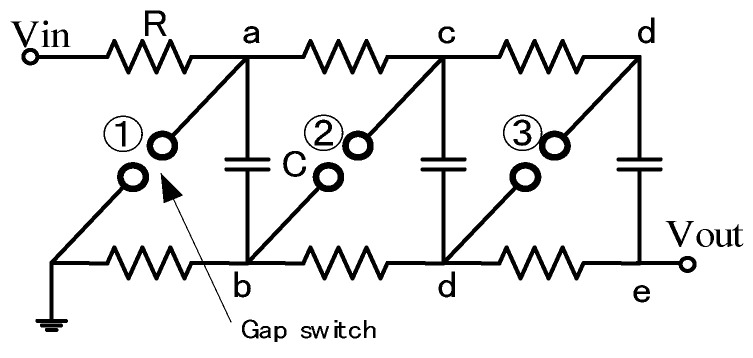
Schematics of a three-stage Marx circuit. *V*_in_: charging voltage, R: charging resistor, C: energy storage capacitor, *V*_out_: output voltage.

**Figure 8 molecules-26-06288-f008:**
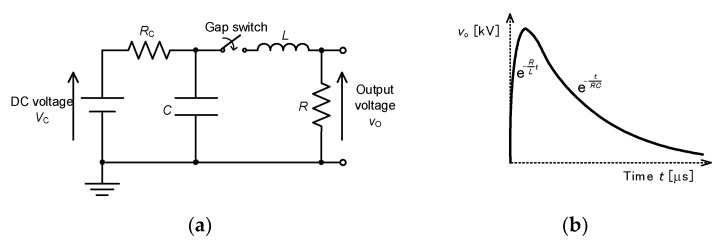
Single-stage impulse generator: (**a**) circuit and (**b**) waveform of output voltage at ${\left(\frac{R}{L}\right)}^{2}-\frac{4}{LC}\gg 0$. *R*_C_: charging resistor, *C*: energy storage capacitor, *L*: circuit inductance, *R*: resistor.

**Figure 9 molecules-26-06288-f009:**
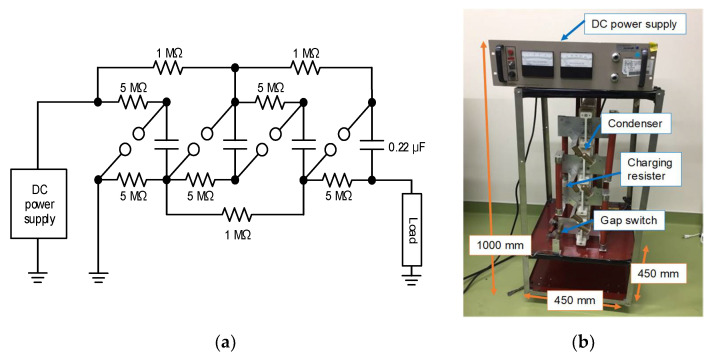
Schematic (**a**) and photograph (**b**) of a four-stage Marx circuit for agricultural applications [23]. © MDPI 2018.

**Figure 10 molecules-26-06288-f010:**
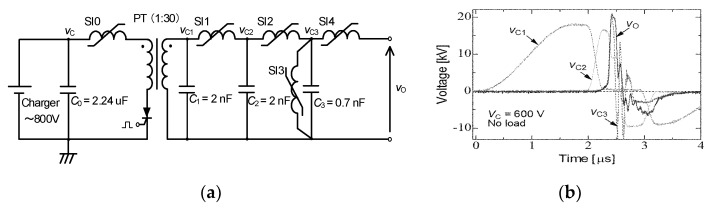
Schematic of (**a**) an MPC circuit and (**b**) its output voltage waveforms. *C*_0_: energy storage capacitor, PT: pulse transformer, *C*_1_, *C*_2_, *C*_3_: secondary capacitors, *SI*_1_, *SI*_2_, *SI*_3_, *SI*_4_: saturable inductors.

**Figure 11 molecules-26-06288-f011:**
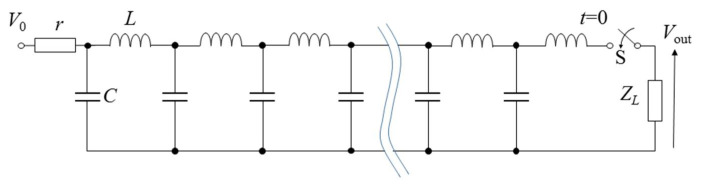
Diagram of a pulse-forming network.

**Figure 12 molecules-26-06288-f012:**
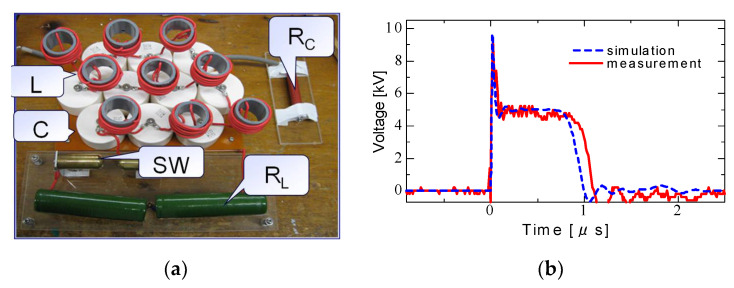
Schematics of a designed PFN (**a**) and its output voltage (**b**) by connecting a load resistance of *Z*_0_ = *Z*_L_ at *C* = 2 nF, *L* = 1.25 mH and *N* = 10.

**Figure 13 molecules-26-06288-f013:**
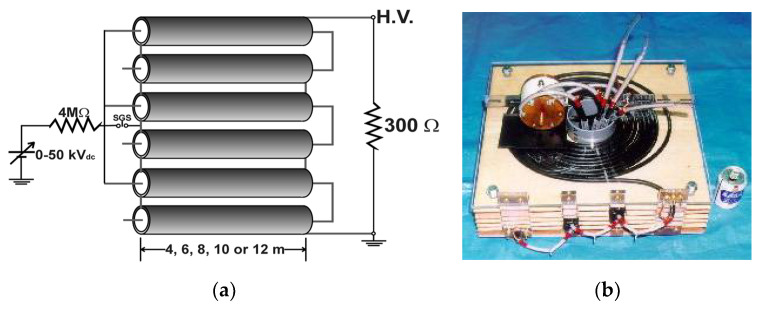
Circuit diagram (**a**) and photograph (**b**) of a three-stacked Blumlein line generator consisting of six coaxial lines [26]. © IEEE 2000. With permission of IEEE.

**Figure 14 molecules-26-06288-f014:**
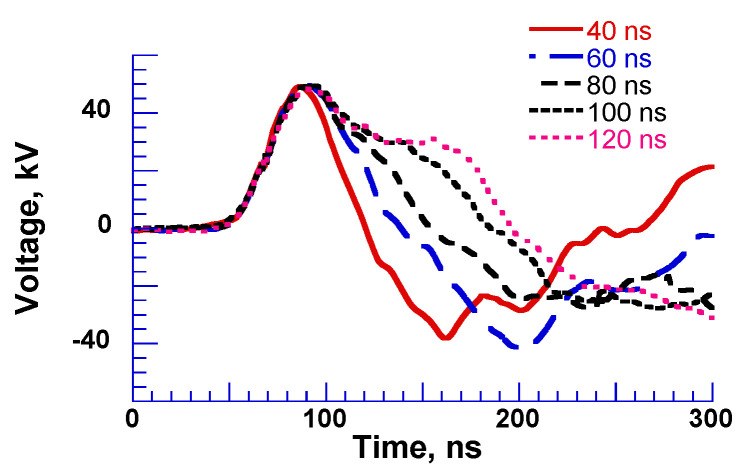
Output voltage of Blumlein line for different lengths of cable with a plasma reactor as a load [26]. © IEEE 2000. With permission of IEEE.

**Figure 15 molecules-26-06288-f015:**
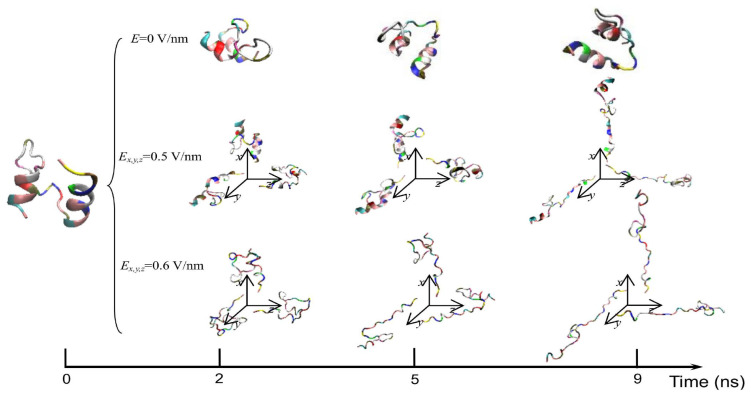
Typical conformations of 1BBL protein exposed by intense electric fields *Ex*, *y* and *z* = 0, 0.5, and 0.6 V/nm [30]. © MDPI 2019.

**Figure 16 molecules-26-06288-f016:**
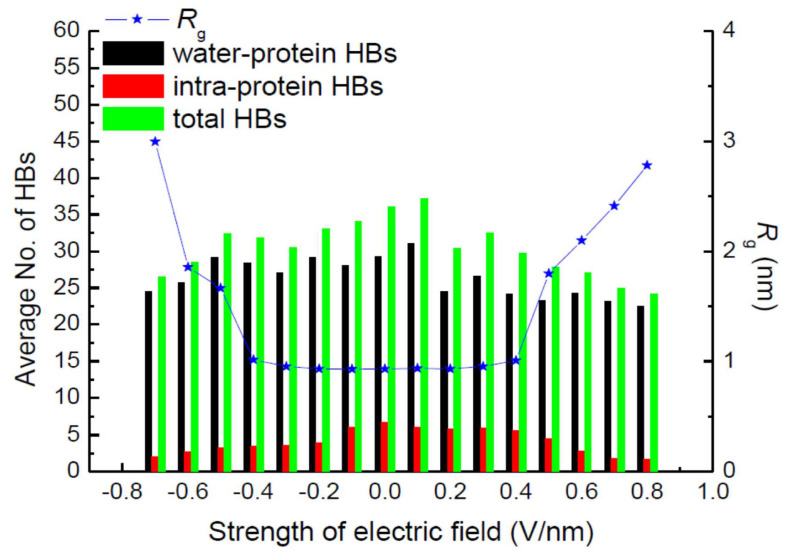
Average number of hydrogen bonds and radius of gyration, *R*_g_, of the 1BBL protein exposed in the electric fields along the z-direction with different strengths [30]. © MDPI 2019.

**Figure 17 molecules-26-06288-f017:**
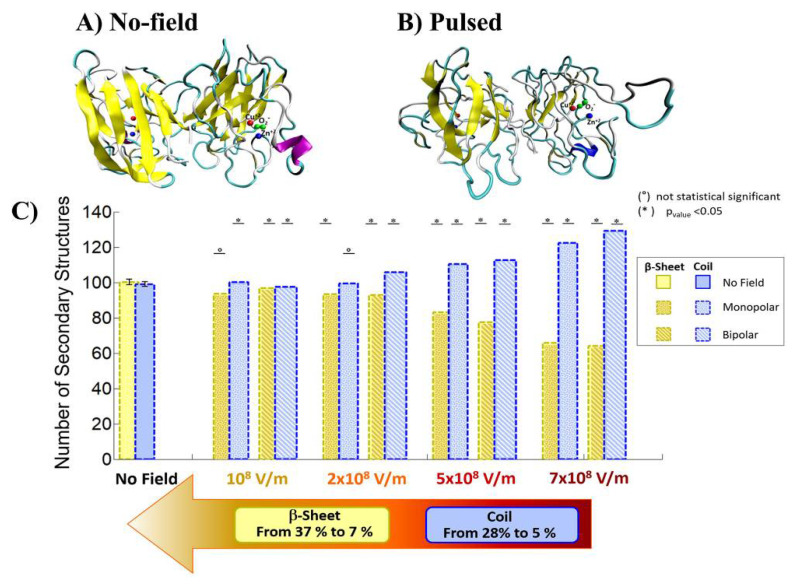
Comparison of the conformation of SOD1 before (**a**) and after an exposure to an electric field of 7 × 10^8^ V/m strength (**b**). Number of coil and β-sheet secondary structures (**c**) [33]. © PLOS 2019.

**Figure 18 molecules-26-06288-f018:**
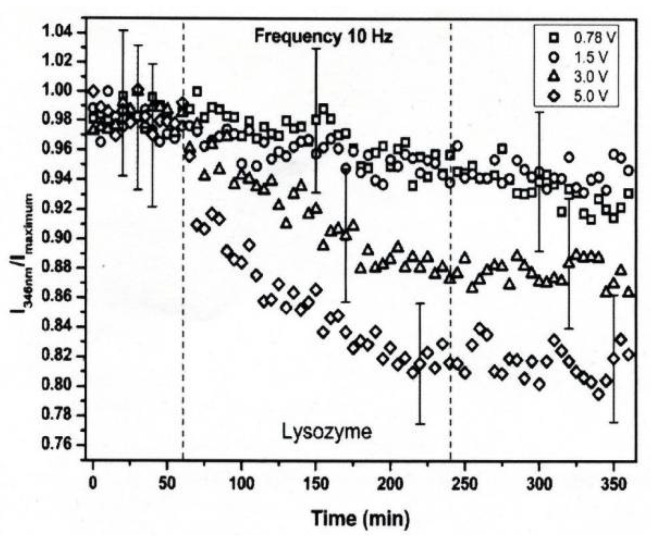
Time-evolution of the relative emission intensity of lysozyme solutions at 0.2 mg/mL (pH 7.2) monitored at 346 nm for exposure to varying electric field strengths. The electric field strengths are (□) 0.78, (○) 1.5, (△) 3.0 and (◇) 5.0 V/cm. The dotted lines indicate partitioning into the first 1 h without exposure to the electric field, followed by 3 h of electric field exposure and a further 2 h of without the electric field [29]. © Royal Society of Chemistry 2014. With permission of Royal Society of Chemistry.

**Figure 19 molecules-26-06288-f019:**
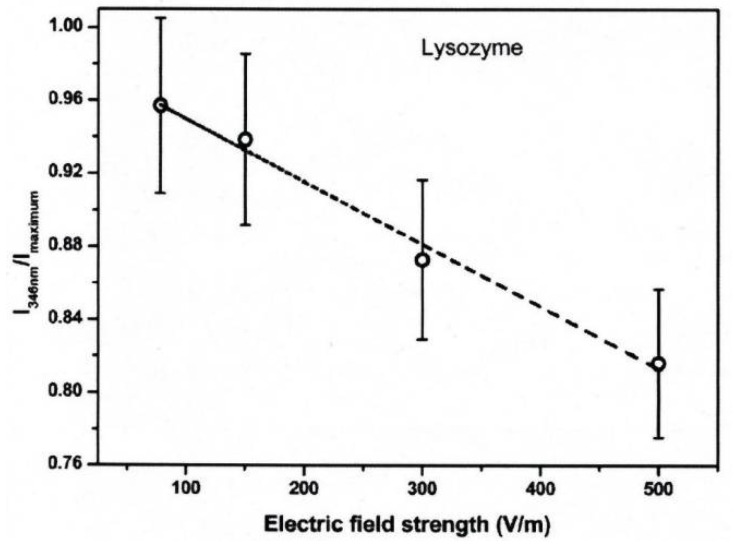
Relative fluorescence emission intensity of lysozyme solutions (0.2 mg/mL, pH 7.2) monitored at 346 nm as a function of electric field strength after 3 h of electric field exposure. The dotted line is a linear fit with an *R*^2^ of 0.99 [29]. © Royal Society of Chemistry 2014. With permission of Royal Society of Chemistry.

**Figure 20 molecules-26-06288-f020:**
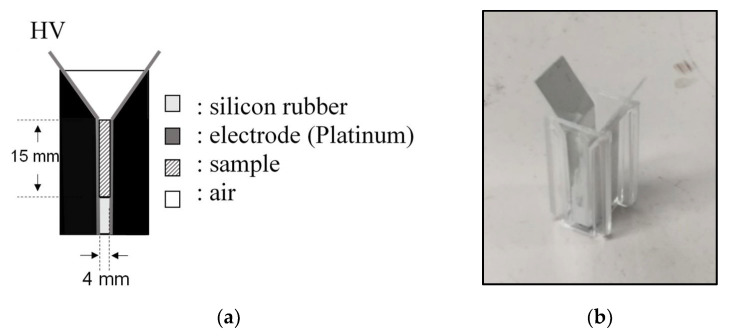
Schematic diagram (**a**) and picture (**b**) of PEF treatment vessel used in a basic experiment of enzyme inactivation [6]. © Elsevier B.V. 2021. With permission of Elsevier B.V.

**Figure 21 molecules-26-06288-f021:**
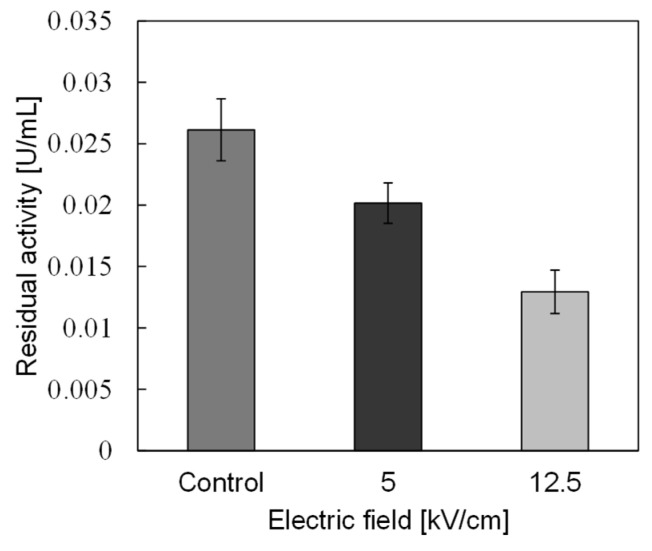
Residual activity of α-amylase for a different strength of PEF at the same input energy of 720 J [9]. © IOP 2021.

**Figure 22 molecules-26-06288-f022:**
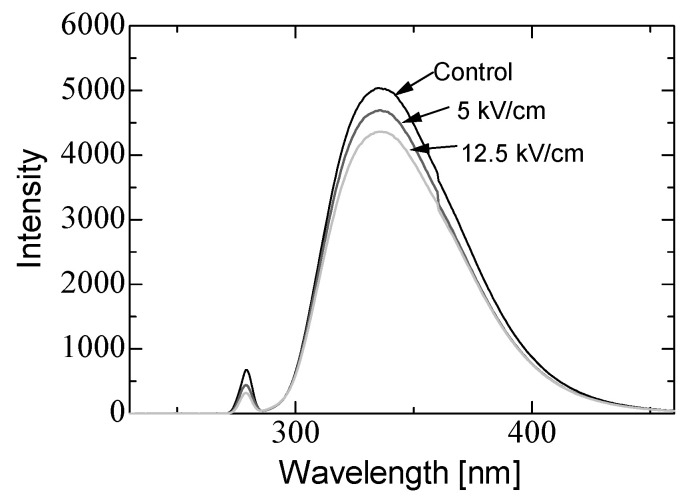
Fluorescence spectra of α-amylase at a 280 nm wavelength of excitation light emission after PEF treatment for various strengths of PEF at the same input energy of 720 J [9]. © IOP 2021.

**Figure 23 molecules-26-06288-f023:**
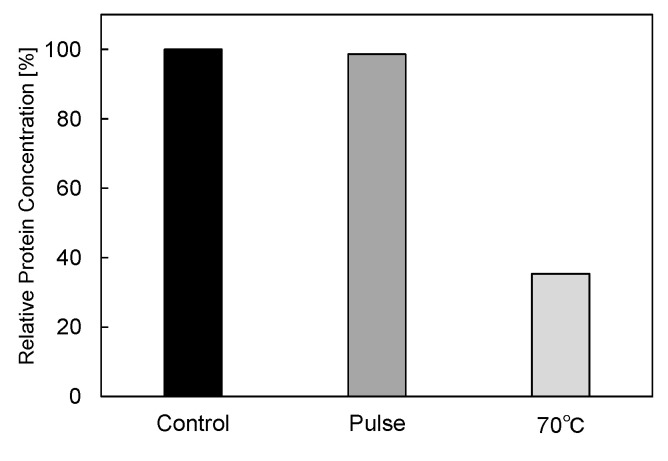
Residual protein concentration after PEF and heat treatments. PEF treatment at 12.5 kV/cm and heat treatment at 70 °C had the same treatment time of 40 min [9]. © IOP 2021.

**Figure 24 molecules-26-06288-f024:**
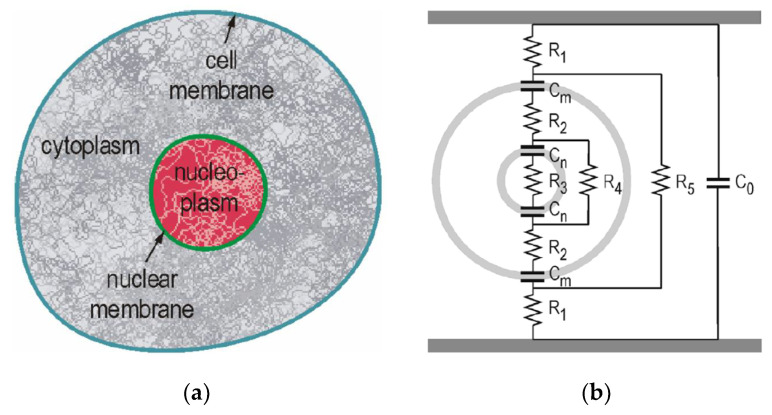
Cross-section schematic of a cell with a nucleus (**a**) and its equivalent circuit (double shell model) in suspension (**b**) [49,50]. © IEEE 2007. With permission of IEEE.

**Figure 25 molecules-26-06288-f025:**
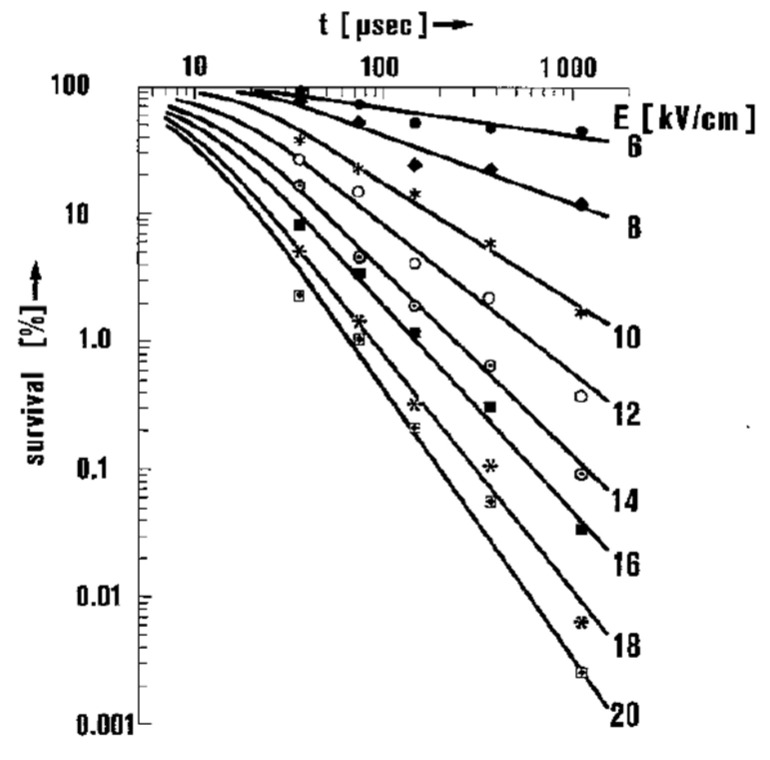
Survival rate of *E. coli* as a function of electric field exposure time for various field strengths [51]. © Springer Nature 1981. With permission of Springer Nature.

**Figure 26 molecules-26-06288-f026:**
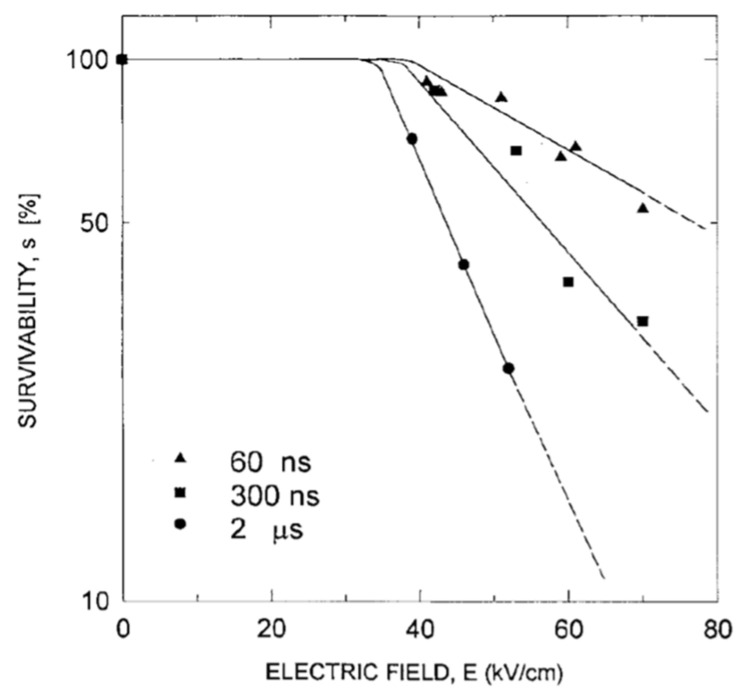
Effect of electric field pulses on the survivability of *E. coli* in tap water for 60 ns, 300 ns, and 2 ms pulse widths [48]. © IEEE 1997. With permission of IEEE.

**Figure 27 molecules-26-06288-f027:**
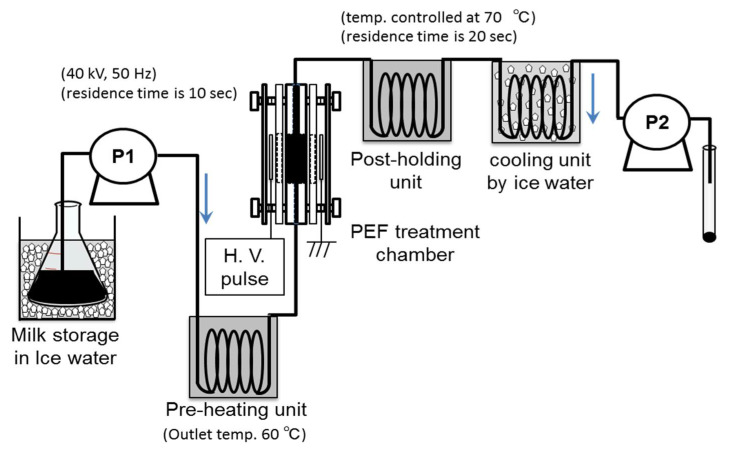
Schematic of a PEF pasteurization apparatus with pre-heating and post-holding units. Reprinted from [59]. © Elsevier Ltd. 2016. With permission of Elsevier Ltd.

**Figure 28 molecules-26-06288-f028:**
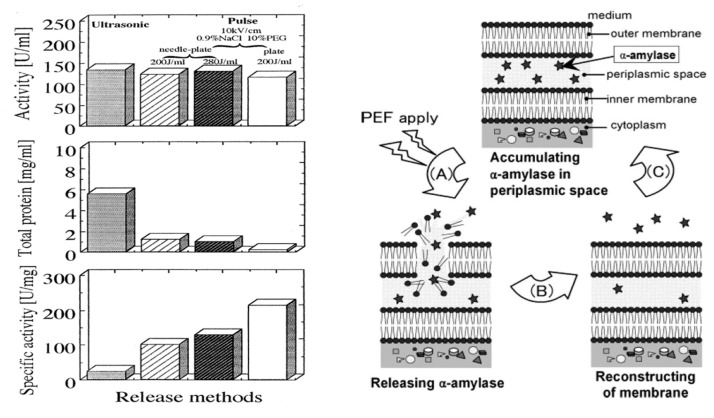
Recombinant *E. coli*/pHI301A and α-amylase activity recovery after PEF extraction of intracellular proteins. Reprinted from [66]. © American Institute of Chemical Engineers 2008. With permission of American Institute of Chemical Engineers.

**Figure 29 molecules-26-06288-f029:**
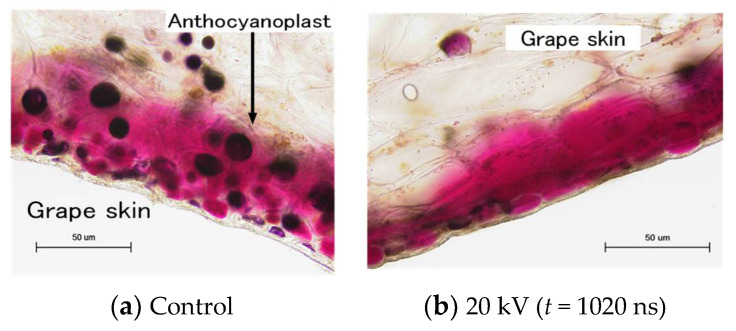
Optical microscopic images of reactions inside the grape skin cell with and without PEF treatment.

**Figure 30 molecules-26-06288-f030:**
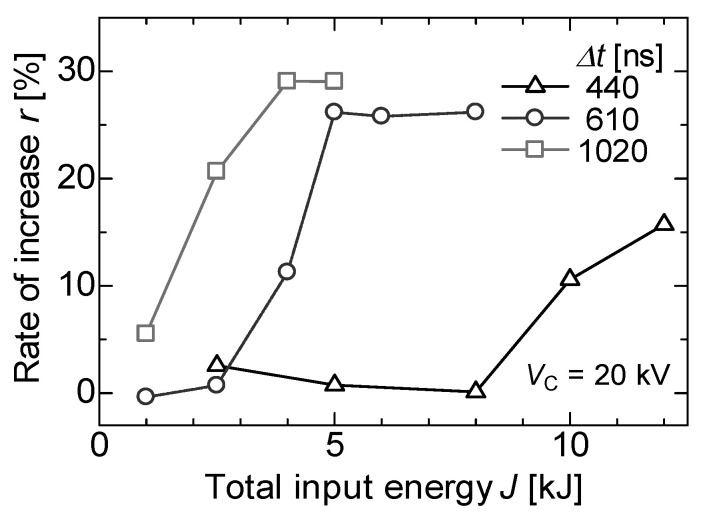
Increased rates of polyphenol extraction by PEF treatment for different pulse widths compared to water extraction at the same temperatures [69]. © The Institute of Electrical Engineers of Japan 2013. With permission of The Institute of Electrical Engineers of Japan.

**Figure 31 molecules-26-06288-f031:**
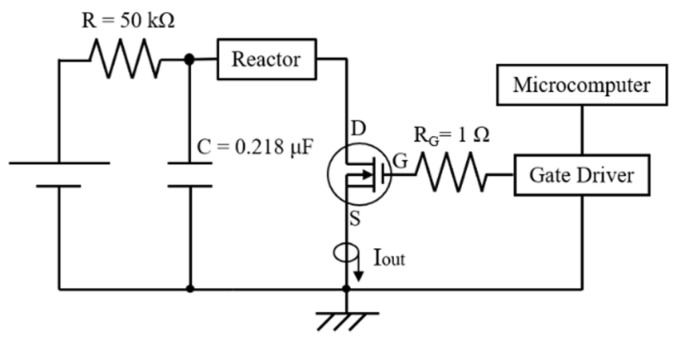
Schematic diagram of the SiC-MOSFET PEF generator for the pre-treatment of drying [84]. © John Wiley & Sons, Inc. 2020. With permission of John Wiley & Sons, Inc.

**Figure 32 molecules-26-06288-f032:**
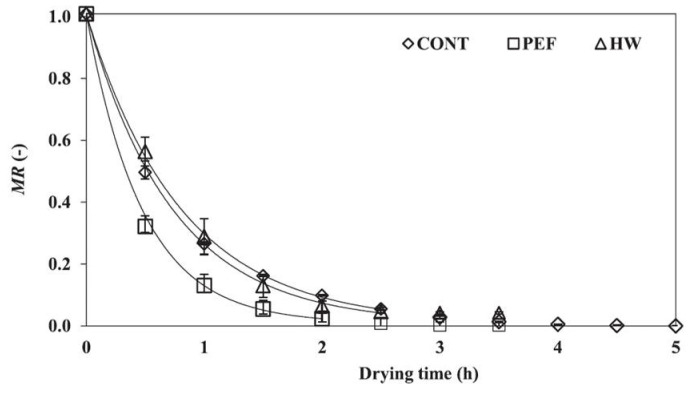
Changes in the moisture content (moisture ratio; MR) of each sample during hot-air drying (*n* = 3). CONT: control; PEF: pulsed electric field; and HW: hot-water pre-treatments [85]. © Elsevier Ltd. 2021. With permission of Elsevier Ltd.

**Figure 33 molecules-26-06288-f033:**
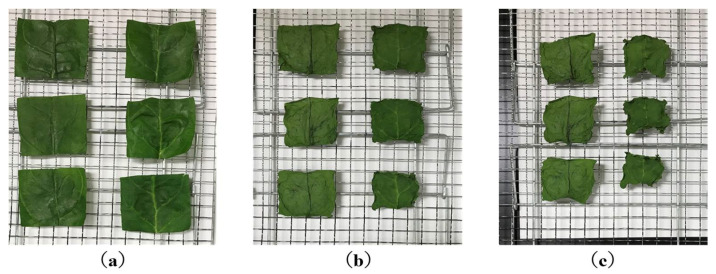
Pictures of sample surfaces before and after drying in hot air. The left side of the samples are the PEF samples and the right side of the samples are the CONT samples. (**a**–**c**) are the mean dried samples after drying for 0, 90, and 210 min, respectively [85]. © Elsevier Ltd. 2021. With permission of Elsevier Ltd.

**Table 1 molecules-26-06288-t001:** The drying constant, *k*, and the ratio of the drying constant of each sample. CONT: control; PEF: pulsed electric field; and HW: hot-water pre-treatments [85]. © Elsevier Ltd. 2021. With permission of Elsevier Ltd.

	Drying Constant *k*	Ratio of Drying Constant
CONT	1.24	-
PEF	2.11	1.70
HW	1.37	1.10

**Table 2 molecules-26-06288-t002:** Ratios of sample surface areas and residual ratios of L-AsA in samples after drying in hot air. CONT: control; PEF: pulsed electric field; and HW: hot-water pre-treatments [85]. Each value represents the means ± S.E. (*n* = 3–7). Different letters (e.g., a and b) are significantly different according to a Tukey–Krammer test (*p* < 0.05). © Elsevier Ltd. 2021. With permission of Elsevier Ltd.

	Ratio of Surface Area (−)	Residual Ratio of L-AsA (−)
CONT	0.352 ± 0.015 a	0.74 ± 0.04 a
REF	0.489 ± 0.002 b	0.74 ± 0.04 a
HW	0.349 ± 0.004 a	0.52 ± 0.03 b

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
