# Peer review of "Pulsed Power Applications for Protein Conformational Change and the Permeabilization of Agricultural Products"

_molecules, 2021, doi:10.3390/molecules26206288_

Round 1

Reviewer 1 Report

The authors present in this paper a review of the pulsed power applications in food processing. The paper deals with the types of the pulsed power systems and their construction in general. The applications to the food processing and their effects are presented thorough the changes in the protein conformation, change in the enzyme activity and effects of the permeabilization. The paper is well organized with main ideas clear to follow. The English is good and with only small mistakes that can easily be corrected. A minor suggestion is to adjust the title in the Section 4. From this title it is not immediately clear to the reader what cell poration has to do with the voltage buildup, antimicrobial activity and pasteurization. Of course that all is true and that when reading full text connection is there, but for someone just entering the field this can be somewhat confusing. I suggest to the authors to adapt the titles or to give a more detail overview of what can be found in Section 4 at the introductory part of this section.

Author Response

Thank you for the valuable comment. I changed the section title from “PEF poration of cell membrane” to “PEF poration process of cell membrane and its applications” to remove the risk of the reader confusion. The changed line is marked by red color.

Reviewer 2 Report

This review paper presents the enzyme activity change and permeabilization of agricultural products using pulsed electric fields generated from pulsed power technologies. Sufficient sources were cited and the analysis presented provides an excellent summary and insights on the subject. Hence, I recommend acceptance after minor revision. Below are few issues that must be addressed by the authors:

  1. Some figures must be fixed since it coincided with the line numberings, some labels on the figures cannot be read because of these.
  2. Define the significance of a and b indicated beside the data in Table 2. What do they signify? 

Author Response

  1. Some figures must be fixed since it coincided with the line numberings, some labels on the figures cannot be read because of these.

Ans.: I am sorry for the confusion. I used the template of molecules for the submission. The template included the line numbers as default. I added the manuscript without line numbers for checking the labels of figures.

  1. Define the significance of a and b indicated beside the data in Table 2. What do they signify?

Ans.: Thank you for the comment. I added the description of the different letters as “Each value represents the means ± S.E. (n = 3–7). Different letters (a, b) are significantly different according to Tukey-Krammer test (P< 0.05)” in the caption. The added lines are marked by red color.
